# “Hang in There!”: Mental Health in a Sample of the Italian Civil Protection Volunteers during the COVID-19 Health Emergency

**DOI:** 10.3390/ijerph18168587

**Published:** 2021-08-14

**Authors:** Rita Roncone, Laura Giusti, Silvia Mammarella, Anna Salza, Valeria Bianchini, Annalina Lombardi, Massimo Prosperocco, Elio Ursini, Valentina Scaletta, Massimo Casacchia

**Affiliations:** 1Department of Life, Health and Environmental Sciences, University of L’Aquila, Via Spennati 1, Edificio Delta 6, Studio 110-Coppito, 67100 L’Aquila, Italy; laura.giusti@univaq.it (L.G.); silvia.mammarella@graduate.univaq.it (S.M.); anna.salza@univaq.it (A.S.); valeria.bianchini@univaq.it (V.B.); massimo.casacchia@univaq.it (M.C.); 2University Unit Rehabilitation Treatment, Early Interventions in Mental Health, Hospital S. Salvatore, 67100 L’Aquila, Italy; 3LARES Italia—Unione Nazionale Laureati Esperti in Protezione Civile, 67100 L’Aquila, Italy; annalina.lombardi@aquila.infn.it (A.L.); massimo.prosperococco@univaq.it (M.P.); elioursini@uniprotezionecivile.it (E.U.); valentinascaletta@uniprotezionecivile.it (V.S.)

**Keywords:** COVID-19, pandemic (COVID-19), Civil Protection, voluntary service, survey, psychological and psychopathological assessment, coping strategies

## Abstract

Few studies have been conducted on civil volunteers and their emotional conditions concerning the current COVID-19 pandemic. The present study aimed to evaluate the impact of the COVID-19 emergency on the mental health (general well-being, depression level, and post-traumatic distress), coping strategies, and training needs in an Italian sample of 331 Civil Protection volunteers of the L’Aquila province, during the first nationwide “lockdown” (8 March–3 June 2020). The rate of respondents to the online survey was limited (11.5%), presumably because displaying distress would be considered a sign of “weakness”, making volunteers unable to do their jobs. More than 90% of the volunteers showed good mental health conditions and a wide utilization of positive coping strategies, with the less experienced displaying better emotional conditions compared to colleagues with 10 or more years of experience. The type of emergency, the relatively few cases of contagion and mortality in the territory compared to the rest of Italy, and the sense of helping the community, together with the awareness of their group identity, could have contributed to the reported well-being. These results may help to identify the needs of volunteers related to this new “urban” emergency to improve both their technical and emotional skills.

## 1. Introduction

In Italy about six and a half million people (12.6% of the population) volunteer in an organized way (7.9%) or individually, based on the Italian National Statistics Institute, ISTAT, data [1]. “Volunteering” contributes to psychological well-being, promotes political participation, generates trust, and also seems to have a professionalizing value [2]. Volunteers represent an important resource in disaster responses [3] and what appears to move them to act seems to be the result of a shared values set and a responsibility culture towards one’s community and society.

However, this is not enough to face emergencies functionally: in addition to these values, the skill of solving problems collectively, experience, training, and knowledge are also needed [4]. This could have been the main reason for the implementation of Civil Protection volunteering, born under the pressure of the great emergencies that have hit Italy in the last 60 years, especially the Florence flood in 1966 and the earthquakes in Friuli (a north-east region) in 1976 and Irpinia (central-southern Italy) in November 1980. A great mobilization of citizens made it clear that what was lacking was not spontaneous solidarity but an organized public system that knew how to use and enhance it.

The Italian Department of Civil Protection (CP) encourages the participation of volunteer organizations in civil activities and emergency responses [5,6,7]. They include psychosocial activities; social welfare activities, as assistance to the most vulnerable people (young people, the elderly, the sick, the disabled); garrison of the territory; logistics, and organizational support in case of natural or human-caused disasters, administrative and secretarial activities; prevention and active fight against forest and interface fires; non-specialist site restoration activities and preparation and administration of meals; driving special vehicles; activities in the field of radio and telecommunications; diving activity; canine activities, etc.). More than one million people throughout Italy and more than 5000 organizations are registered on the list of the National Civil Protection Department.

The Italian Civil Protection system has dealt with several disasters. The largest one was the 2009 earthquake of magnitude 6.3 that hit the province of L’Aquila claiming the lives of 309 people, injuring thousands of citizens, causing tens of thousands of displaced people, and provoking severe material destruction [7,8,9]. In a few hours, the Fire Brigade, the voluntary organizations of the Civil Protection National Service, and the Army were deployed in L’Aquila from all over Italy to search for and rescue the victims of the earthquake. On the occasion of the L’Aquila earthquake, the authors could appreciate the work of Civil Protection and cooperated with them in the following months [10,11,12], in a difficult context that has seen a growth in the awareness of the population in community aid and voluntary activities.

In a post-disaster earthquake setting, the health workers [13,14] and also the volunteers could be exposed to the risks of burnout, anxiety, depression, and post-traumatic stress disorder (PTSD) [15,16,17]. PTSD requires the diagnostic criterion of (A) exposure to a traumatic or stressful event as a diagnostic criterion [18]. PTSD is characterized by four distinct symptom clusters: (B) intrusion symptoms, i.e., re-experiencing of the traumatic event through such phenomena as dreams, flashbacks, and intrusive, distressing thoughts; (C) avoidance and numbing, characterized by avoidance of trauma reminders and numbing of emotions; (D) negative alterations in cognitions and mood, characterized by negative thoughts or feelings that began or worsened after the trauma, and (E) hyperarousal, characterized by difficulties sleeping and concentrating, irritability, and hypervigilance. The duration of the disturbance is more than 1 month (F), and symptoms create distress or functional impairment (e.g., social, occupational) (G). The prevalence of PTSD among direct victims has been reported to be 30–40% while among rescue workers it was 10–20% [19]. In the first month after the 2009 L’Aquila earthquake, two female health volunteers (1.6%), although not exposed directly to the catastrophic event, were among the 122 help-seekers at the San Salvatore General Hospital Psychiatric Unit, affected by serious acute stress disorder [8].

However, research has found that potentially traumatic experiences do not necessarily lead to the development of psychopathological symptoms [20].

Many variables influence and determine the type of reaction that the individual manifests as a result of having operated in stressful emergency contexts [21,22,23,24]. These include personal and environmental factors that could be considered either risk or protective factors for the volunteers.

Regarding personal factors, considerable attention has been paid to the aspect of individual resilience and the coping strategies adopted following trauma [25]. Positive coping strategies can be defined as behavioural and psychological efforts employed to overcome, tolerate or reduce the impact of stressful events and promote positive psychological outcomes. They act as protective factor against the development of post-traumatic stress and emotional distress symptoms, representing important resilience factors for the individual [26,27,28,29]. Emotional distress symptoms include physical symptoms, such as headaches, shortness of breath, feeling tired, sleeping disorders, and mental or behavioral symptoms, such as being more emotional than usual, feeling overwhelmed or on edge, trouble keeping track of things or remembering, trouble making decisions, solving problems, concentrating, getting work done. Individuals will be oriented toward better outcomes and predisposed to post-traumatic growth (PTG), characterized by an improvement in relationships with others, and a change in the vision they have of themselves and life [30,31]. Furthermore, various studies report how personality factors, such as extroversion, open-mindedness, sympathy, conscientiousness, self-efficacy, and optimism, allow rescuers to experience growth when associated with the right coping skills [32].

With regard to the environmental factors that influence the way to react to an emergency, it has been found how the ability to resolve conflicts or potential conflicts within the organization, improving the interest levels of volunteers, the implementation of recruitment and retention policies for volunteer into the team, adequate education and training, promoting social interaction skills, strengthening teamwork and building an effective team can make a difference [33,34]. It has been highlighted that often volunteers, compared to professional rescuers, have lower levels of training and resources available; consequently, they are more exposed to the risk of experiencing problems related to stress [35,36]. The type of emergency, level of exposure, working conditions, leadership style, perceived social support, and the presence of any psychological support are also important environmental factors [37]. Teamwork and a strong sense of community are key protective factors for rescue workers in the event of a disaster [38].

So far, the knowledge of rescue workers comes from catastrophic events with rapid onset and triggered by natural events; very little is known about the nature and forms of “urban emergencies” in long-standing crises, such as in contexts characterized by governance failures, conflicts (political, social, ethnic), violence, crime [39] or relative to a global pandemic, such as COVID-19. From the analysis of the literature on intervention in pandemics, the majority of the studies were related to health professionals, with a strong psychological impact on the staff involved [40], as was observed in operators during the severe acute respiratory syndrome (SARS) epidemic in 2003 and the Ebola virus in 2013 [41].

Italy was one of the first countries to experience a major epidemic of severe acute respiratory syndrome coronavirus 2 (SARS-CoV-2), with >1000 cases confirmed by 1 March 2020 [42]. In the first wave of SARS-CoV-2, the majority of cases occurred in northern Italy in the region of Lombardy.

In the Abruzzo—a central Italy region; with 1,293,941 inhabitants, data ISTAT 2009 [43]—the first infection from coronavirus was diagnosed on 27 February 2020. It was a man from Brianza, who arrived in Roseto (Teramo) with his family and, at the onset of the first symptoms, was transported to the Mazzini hospital in Teramo. Just two days later, on 29 February 2020, the first person infected in the province of Chieti was hospitalized in Pescara after returning from a business trip to Lombardy. The next day, the confirmation came from the Spallanzani institute in Rome: it was coronavirus.

On 5 March, 8 subjects infected with COVID-19 were diagnosed in Abruzzo. A few days later, on 9 March, there were 33. On 29 May, a few weeks after the end of the “first Italian lockdown” and the recovery of a new normal daily life in Abruzzo, no new infection was reported, until the “double zero” (no new cases, and no deaths) arrived on 8 June 2020.

As of 3 June 2020, in Italy, 233,836 COVID-19 cases had been confirmed and a total of 33,602 people had died; Lombardy (10,027,602 inhabitants) registered 89,442 cases (0.89% of the regional population) and 16,172 (0.16%) deaths. In Abruzzo, a total of 3252 people had tested positive for COVID-19 (0.25% of the total population) and 414 people had died (0.03%) [44], representing <2% of cases in Italy. During the first COVID-19 wave, compared to other Italian regions, Abruzzo was less hit and the province of L’Aquila was less hit compared to the Adriatic Sea areas (in particular, the town of Pescara) of the same region.

During the COVID-19 pandemic, both internationally and nationally, it was seen right from the start (March 2020) that it was necessary to involve a large number of operators and volunteers to deal with the spread of an infectious disease of such magnitude. For example, data from a study carried out in China, already in the first months of the COVID-19 pandemic, showed that the role of volunteers was crucial, and more than 85,699 rescue workers were recruited through mobile apps [45].

In Italy, the recruitment of volunteers in the COVID-19 emergency was fundamental to ensure support for the territory in terms of logistical aspects and assistance to the population [29,46]. The psychological impact of this emergency on volunteers highlighted the protective and mediating function of positive coping strategies on the incidence of stress in secondary trauma [27,28], and high levels of burnout syndrome correlated to coping strategies, as denying and minimizing the threat of the stressful events [29].

This study aimed to evaluate the impact on the mental health of the emergency caused by COVID-19 in an Italian sample of 331 Civil Protection volunteers of the L’Aquila province territory, during the first nationwide “lockdown” (8 March–3 June 2020). The aims of the study were to (1) describe the psychopathological health conditions, and the different (problem-focused, emotion-focused and dysfunctional) coping styles adopted by the volunteers in the sample; (2) examine the correlations among emotional distress, psychopathological symptoms, and individual coping strategies, and years of volunteers’ service; and (3) assess the training needs of the volunteers.

We hypothesized that volunteers serving during the COVID-19 pandemic could be more distressed compared to non-serving volunteers using: (1) more dysfunctional; (2) fewer problem-focused and (3) fewer emotion-focused strategies. We also hypothesized that a longer experience in volunteers service could be associated with better mental health conditions and positively correlated with functional coping strategies. Furthermore, we expected a large request of training courses and/or workshops and/or support sessions to better manage distress/anxiety by volunteers, to improve their skills to face distressing emergencies.

## 2. Materials and Methods

### 2.1. Study Design and Participants

The study was conducted following the agreement signed on 23 June 2020, between the research group coordinated by R.R., belonging to the University of L’Aquila, and the voluntary associations of the Regional Civil Protection system: the Civil Emergency Voluntary Intervention (PIVEC) of L’Aquila, and the Civil Protection Nucleo Operativo Volontari (NOVPC) of Tagliacozzo (L’Aquila). The agreement was strongly supported by the Abruzzo section of LARES (National Union of Graduated Experts in Civil Protection), born from the collaboration between CETEMPS, Geolab (UNIVAQ), and LARES Italia. The LARES Italia association recruits students and graduates who are experts in the Civil Protection sector who already have adequate training in their professional culture that makes them capable and flexible in dealing with various emergencies. The agreement aimed, first, to assess the psychological and psychopathological conditions of the volunteers often subjected to severe stress without the possibility of being able to express their feelings. The second aim of the agreement was the investigation of training needs with a view to organizing training courses.

We conducted a cross-sectional online anonymous survey design using a convenience sample in the period 15 July–30 August 2020, referred to as the first Italian nationwide “lockdown” (8 March–3 June 2020).

The survey represented the results of one online focus group planned on Microsoft Teams^®^ (Microsoft Corporation, Redmond, WA, USA) to develop concepts and questions for the questionnaire design. The focus group meeting lasted 2 h and included all the authors of this work.

On 15 July 2020, the survey was addressed through the Civil Protection associations to 290 PIVEC and 41 NOVPC volunteers, who received an invitation to participate in the survey by e-mail through a link to a “Google Module” form. The participants did not receive any form of compensation for participation in this study. The data were collected up to the end of August 2020.

### 2.2. Survey Instrument and Related Measures

Google Forms^®^ was used to create the online survey. Three main sections were included in this anonymous online survey.

The first preliminary section concerned information on the study and the protection of privacy, ensuring total confidentiality of the personal data as also provided by the Italian Legislative Decree 10 August 2018, no. 181, provision for the adaptation of national legislation to the European General Data Protection Regulation n. 2016/679. A lack of consent would not allow the compilation of the form.

The second section concerned information on socio-demographic characteristics, qualifications and training courses, previous experience in the field of emergencies with specific questions on the level of involvement during the COVID-19 emergency, and the current needs for psychological support and training (Table 1).

The third section included instruments to investigate anxious and depressive symptomatology, post-traumatic distress, and the adopted coping styles. In the end, the volunteers were asked the time needed to fill in the online survey.

#### 2.2.1. Anxious and Depressive Symptomatology

The 12-Item General Health Questionnaire (GHQ-12) [47,48,49] is the most extensively used screening instrument for common mental disorders, in addition to being a more general measure of psychiatric well-being. The GHQ-12 consists of 12 items, each one assessing the severity of a mental problem over the past few weeks using a 4-point Likert-type scale (from 0 to 3). The score was used to generate a total score ranging from 0 to 36. High scores indicated poor health. The scores fell into three categories: 0–14 = normal range, 15–19 = moderate psychological distress, and 20–36 = severe psychological distress.Internal consistency for the GHQ-12 was high in this sample (Cronbach’s α = 0.829). The Patient Health Questionnaire-9 (PHQ-9) is used for the diagnosis, monitoring and study of the severity of depression in primary care [50]. It is composed of 9 items corresponding to the symptoms of major depression according to DSM 5. The severity level of depression is broken down by PHQ-9 scores: 1–4 = no depressive symptoms; 5–9 = minimal depressive symptoms/subthreshold depression; 10–14 = minor depression/mild major depression; 15–19 = moderate major depression; and ≥20 = severe major depression. A score of 10 is indicated as the point at which the sensitivity and specificity of the instrument are recognized as optimal for highlighting depressions of clinical relevance [51]. In this sample the internal consistency for the PHQ-9 was high (Cronbach’s α = 0.865).

#### 2.2.2. Traumatic Distress

The Post-Traumatic Stress Disorder Checklist for DSM-5 (PCL-5), developed in 1990 by the National Center for PTSD in Washington DC, is a tool used for the provisional diagnosis, screening, and follow-up of PTSD symptoms [52]. The PCL-5 is composed of 20 items that are used for the evaluation of the corresponding symptoms of PTSD exposed in the DSM-5 (intrusiveness, avoidance, negative cognitions and mood, and hyperarousal). A score from 0 (not at all) to 4 (very much) is assigned to each item that refers to symptoms that have created a disturbance in the last month. Scores consist of a total symptom severity score (from 0 to 80). A score between 31 and 33 is indicative of the probable presence of PTSD. The internal consistency for the PCL-5 was excellent (Cronbach’s α = 0.937).

#### 2.2.3. The Coping Styles

The Coping Orientation to Problem Experienced (Brief COPE) is a self-report tool [53] that was developed in its short version of 28 items starting from the original 60 items. The scale aims to evaluate the characteristics of the coping of subjects as normal or suffering from different pathologies, e.g., somatic and psychic. It was also not designed for repeated evaluations, although its administration can be repeated at various intervals of time. It includes 28 items divided into 14 scales (each of 2 items): (1) positive reframing; (2) self-distraction; (3) venting; (4) use of instrumental support; (5) active coping; (6) denial; (7) religion; (8) humor; (9) behavioral disengagement; (10) use of emotional support; (11) substance use; (12) acceptance; (13) planning; (14) self-blame. The items are rated on a 4-point scale, from 1 (usually I don’t do this at all) to 4 (usually I do just that), and the 14 scales present range scores of 2–8. Three composite sub-scales, or dimensions, measuring problem-focused strategies, emotion-focused strategies, and dysfunctional coping strategies have proved useful in clinical research and have content validity [54]. The “problem-focused strategies” consist in the attempt to modify or resolve the situation that is threatening or harming the individual; the “emotion-focused strategies” consist in the regulation of negative emotional reactions resulting from the stressful situation; the “dysfunctional coping strategies” consist of inactive strategies that lead to increased stress [55]. In the present study, the “problem-focused strategies,” identified by 3 scales (use of instrumental support; active coping; planning), the “emotion-focused strategies” identified by 5 scales (positive reframing; religion; humor; use of emotional support; acceptance) and the “dysfunctional coping strategies”, identified by 6 scales (self-distraction; venting; denial; behavioral disengagement; substance use; self-blame), were considered for the data analysis. The internal consistency for the Brief COPE was high (Cronbach’s α = 0.881).

## 3. Statistical Analysis

A descriptive analysis of the sample concerning socio-demographic data was performed. The chi-square test was used to verify the difference between observed frequencies. We used non-parametric tests, in consideration of the very limited and unbalanced sample, in terms of frequency of volunteers engaged during the COVID-19 pandemic (*n* = 30) or not engaged (*n* = 8) and frequency of volunteers less (*n* = 22) or more experienced (*n* = 8) serving during the pandemic. The Mann–Whitney U test was used to verify, in the presence of ordinal values coming from a continuous distribution, if the two groups of volunteers came from the same population. Partial correlation analysis was performed controlling for the effect of age to assess the possible association among the psychological and psychopathological variables and the coping strategies measured in the sample and years of volunteer’s service. The statistical analysis was performed using SPSS version 26. The significance level was set at 0.05.

## 4. Results

Out of 331 Civil Protection volunteers involved, only 38 subjects answered the online survey. The rate of respondents was rather limited (11.5%) due to several reasons, which will be hypothesized and discussed in the following paragraphs. The 30 volunteers (79% of the sample, 23 men, and 7 women) operating during the health emergency were compared to 6 men and 2 women not serving during the COVID-19 pandemic. The mean time to fill in the online survey was reported of 25 min.

### 4.1. Socio-Demographic Characteristics of the Sample and Civil Protection Volunteers Experience

The entire sample included mainly young people (mean age = 37.5 ± 14.1), within a range of 16–67 years; more than three-quarters of the samples were males (*n* = 29, 76.3%), with a medium average level of education.

No statistically significant difference was found in age (men *MdN* = 17.9; women *MdN* = 24.3; *U* (N_men_ = 29 U, N_women_ = 9) = 86.5; z = −1.512; *p* = 0.133), and level of education by gender (chi-square test = 5.693; d.f. 4; *p* = 0.223). For the entire sample, a statistically significant difference by gender was found in the larger percentage of married women (55.6%) compared to men (37.9%), for the prevalence of the latter celibates (55.2%) (chi-square = 8.242; d.f. 3; *p* = 0.041). Regarding working conditions, a higher proportion of men were engaged in full-time activities (employers, teachers, pizza chefs, craftsmen, military men, prison policemen, workers) (58.6%) compared to women (researchers, free-lance professionals, employers) (33.3%). No female students were included in the sample (chi-square = 15.132; d.f. 4; *p* = 0.004). No statistically significant difference was found between the two sub-groups of volunteers in the variables reported in Table 2.

The data regarding one’s experience in the Civil Protection voluntary sector are reported in Table 3.

For one-third of the sample the COVID-19 emergency was their first experience of volunteering in the Civil Protection, whereas more than 20% of the total sample had served for more than 10 years, mainly in the regional and national territory. They experienced service in earthquakes (April 2009 region Abruzzo, May 2012 region Emilia-Romagna, August–October 2016 Amatrice (Rieti) and Central Italy, and January 2017 region Abruzzo and Central Italy), fires (L’Aquila and province territories), avalanches (Rigopiano (Pescara), 18 January 2017), and floods.

No statistically significant difference was found in age between the volunteers with 10 or more years of experience (*Mdn* = 23.5) compared to less experienced volunteers (*Mdn* = 18.2) (U (N_less experienced_ = 29 U, N_more experienced_ = 9) = 94.5; z = −1.237; *p* = 0.221). No statistically significant difference was found in years of volunteering experience by gender (men *MdN* = 20.7; women *MdN* = 15.6; U (N_men_ = 29 U, N_women_ = 9) = 95.5; z = −1.218; *p* = 0.234). During the COVID-19 pandemic, half of the sample was involved in service at an organization level and half in “on the field” interventions. Almost 70% of the volunteers involved during the COVID-19 pandemic reported that volunteering interfered with their work and/or study activities at different levels of severity, with 25% reporting many or major difficulties.

### 4.2. Anxious and Depressive Symptomatology

For the entire sample, the mean total score of GHQ-12 was 8.36 (SD 4.6); 34 subjects (89.5%) rated scores in the normal range. Only one female volunteer (2.6% of the total sample) serving during the COVID-19 pandemic was rated in the range of severe distress. No statistically significant difference was found between the two sub-groups of volunteers in the GHQ-12 scores (Table 4).

The total PHQ-9 score mean was 3.92 (SD 3.4) for the entire sample; 23.7% of the volunteers had subthreshold depressive symptoms, while 7.9% had mild depressive symptoms (two women and one man). Compared to men, women showed a greater impact of depressive symptoms, with a statistically significant difference compared to men (chi-square = 10.739; d.f. 2; *p* = 0.005) (Figure 1). No statistically significant difference was found between the two sub-groups of volunteers in the PHQ-9 scores (Table 4).

A statistically significant difference emerged in the comparison between subjects with volunteer experience longer than 10 years reporting higher GHQ-12 median scores (effect size = 0.12) and displaying a worse emotional condition compared to the less experienced volunteers (Table 5).

### 4.3. Traumatic Distress

More than 90% of the Civil Protection volunteers did not report traumatic distress for the COVID-19 health emergency, as measured by the PCL-5 total score. Only three subjects (7.8%, two women, and one man, all belonging to the serving volunteers) presented a scoring higher than 31 (cut-point score 31–33), indicative of possible post-traumatic symptomatology. No statistically significant differences were found in the PCL-5 scores between the volunteers serving or non-serving during the COVID-19 pandemic (Table 4) neither in the comparison between more experienced and less experienced volunteers (Table 5). The PCL-5 total score did not show statistically significant differences by gender.

### 4.4. Coping Strategies

The problem-facing strategies adopted by the entire sample, as assessed by the Brief-COPE, are reported in Figure 2. Higher scores were referred to the dimensions of “problem-focused” strategies (mean = 5.39, SD = 1.23) and “emotional-focused” strategies (mean = 4.31, SD = 1.01), compared to the dimension of “dysfunctional coping strategies” (mean = 3.05, SD = 0.84), showing, as expected, the larger utilization of effective coping strategies of the subjects included in the study. There were no statistically significant gender differences. No statistically significant difference was found between the two sub-groups of volunteers (volunteers serving during the COVID-19 pandemic vs volunteers non-serving) in the 4 dimensions of Brief-Cope (Table 4). No statistically significant difference was found between the more experienced volunteers and the less experienced in the utilization of coping strategies (Table 5). The volunteers serving during the COVID-19 pandemic showed statistically significant higher scores in the Brief-Cope subscales of “Acceptance” (effect size = 0.11) and “Active Coping” (effect size = 0.10) compared to their colleagues non-serving during the health emergency (Table 6). No statistically significant difference was found between the more experienced volunteers and the less experienced in the subscale scores (Table 7).

### 4.5. Training Needs

The second section of our survey instrument investigated the training for preparedness in the field of emergencies and the need for psychological support and training. Having carried out certified training courses relating to rescue activities was reported by more the 70% of the sample (Table 2). Concerning the estimated need for psychological support, only one-third of the sample reported being willing to share his/her experience with a professional, while approximately 60% of volunteers serving during the COVID-19 pandemic reported preferring to talk about it with friends and colleagues. More than 80% of the sample showed a general interest in improving their skills, mainly following “technical” courses related to operating in disasters. Less than 20% of the sample displayed interest in training to better manage distress/anxiety (Table 2).

### 4.6. Correlation Analysis

Partial correlation analyses controlling for the effect of age were conducted among the total scores of GHQ-12, PHQ-9, the 4 dimensions (B-E criteria of DSM-5) of the PCL5, and the 3 Brief-Cope coping styles identified by “problem-focused strategies”, “emotion-focused strategies”, “dysfunctional coping strategies” and years of volunteers’ service (Table 8). The analyses conducted on the volunteers serving during the COVID-19 pandemic (*n* = 30) showed a strong positive and statistically significant correlation between the total scores of GHQ-12, which indicates the presence of emotional distress, and the total scores of PHQ-9, which identify the presence of depressive symptoms.

A positive correlation was identified between the scores of the PHQ-9 total scores and the 4 dimensions of PCL-5, and, similarly, a positive and statistically significant correlation between the GHQ-12 and three dimensions of PCL-5 (Intrusiveness, Negative cognitions and mood, Hyperarousal) was found. The Brief-Cope dimension of “dysfunctional coping strategies” showed a statistically significant positive correlation with all the emotional distress and psychopathological measures (GHQ-12, PHQ-9 and the four dimensions of the PCL-5), while the dimension of emotion-focused coping strategies was positively correlated with three (Intrusiveness, Avoidance, Negative cognitions and mood) out of the four diagnostic criteria measured by the PCL-5. In addition, the “emotion-focused strategies” dimension was positively correlated with the other two dimensions of the Brief-Cope scale.

No statistically significant correlations between measured variables and years of volunteers’ service were found.

## 5. Discussion

The study aimed to evaluate the impact of the COVID-19 health emergency on the mental health of a sample of Civil Protection volunteers in the L’Aquila area, investigating their coping strategies and their training needs. In addition, the study aimed to analyze the correlations among emotional distress, psychopathological symptoms, and individual coping strategies, and years of volunteers’ service.

The results showed good mental health conditions and a wide utilization of positive coping strategies, in a young population sample of predominantly male volunteers. Despite the impact on work or study activities in more than half of the sample, more than 90% of volunteers participating in the survey did not report a clinical psychopathological impairment. Women, a quarter of the sample, showed a high level of mild depressive symptoms, compared to men, confirming their greater vulnerability to depressive symptoms [56]. Volunteers with more than 10 years of experience displayed worse conditions compared to the less experienced, although with limited clinical meaning. No post-traumatic distress related to COVID-19 pandemic volunteering activities was displayed, with a relatively low rate of approximately 8% of subjects scoring positive for post-traumatic symptomatology, associated with previous personal traumas (complicated grief for the sudden loss of significant others).

Most of the volunteers in our sample showed the use of strategies focused on planning and dealing with problems operationally, making use of adequate instrumental supports. Not surprisingly, in our study, the dimension of “dysfunctional coping strategies” was positively correlated with all the psychopathological variables, evidencing that a maladaptive answer to stressful events can promote “learned helplessness”, and thus depressive symptoms. This result is broadly in line with the literature that recalls how maladaptive strategies such as avoidance and self-blame lead to clearly worse outcomes in terms of psychological well-being [26]. Despite our hypothesis, the volunteers made a modest request for professional emotional support and training courses to better manage distress/anxiety to improve their skills to face distressing emergencies.

Disproving our hypothesis and the findings of Vujanovic et al. [46], the volunteers in our sample serving during the COVID-19 pandemic did not show significant clinical impairment at the psychopathological level, and showed greater use of coping strategies, such as acceptance of the reality and active coping with problems, compared to the non-serving volunteers. They displayed a high utilization of positive-focused strategies and, we can infer, a strong motivation commitment toward their community and their territories, which were seriously hit a little more than 10 years ago by the devastating natural disaster of the 2009 L’Aquila earthquake. The younger people in the volunteering role reported a better emotional condition compared to their more experienced colleagues; this was an unexpected finding, as we estimated that longer experience could represent a source of problem-solving strategies to cope practically and emotionally in emergencies.

Indeed, the COVID-19 pandemic emergency has some different characteristics compared to the more usual type of emergency faced by Civil Protection. Among volunteers not at their first experience, more than 80% served as rescuers in earthquakes, being exposed to unfavorable working conditions that often do not allow the ability to satisfy essential needs (hours and quality of sleep, availability of food, etc.), and in the absence of social support or contact with family members [37]. The previous experiences of our volunteers were characterized by the strong “urgency” in interventions and difficult working conditions, giving a sense of pressure and unsafety. The principal activities carried out by our sample of serving volunteers during the COVID-19 pandemic mainly concerned organizational, secretarial, and monitoring interventions in the offices, with some activities “on the field” (driving ambulances, transporting food, distributing individual protection disposals). A recent Italian study confirms that during the COVID-19 pandemic emergency workers experienced lower levels of stress than nurses and physicians, representing the direct contact with COVID-19 patients one of the risk factors for emergency stress, together with female sex, unexpected events, and lack of personal protective equipment [27].

Concerning the well-being of our volunteers, we could hypothesize, first, that the greatest danger related to contagion, a distinctive feature of emergencies related to pandemics, could have been less perceived in conducting these activities, and considered a “lower risk” compared to the “rescue activities” of natural disasters or the “front-line” clinical hospital ward activities with infected subjects and dying people. Second, we have to contextualize the emergency in our province, less hit in the region of Abruzzo during the first COVID-19 wave, with relatively few cases of contagion and mortality compared to the rest of Italy. Third, the rate of respondents could be biased, and only the more performant volunteers may have answered the online assessment. Fourth, and last, but perhaps most important, our serving Civil Protection volunteers displayed a positive, strong sense of help, together with the awareness of their group identity that enables all members to experience a sense of well-being, belonging, fellowship and cohesion. Our findings are not surprising and confirmed recent studies that found that volunteers serving during the COVID-19 pandemic, although initially experiencing distress conditions, were “happy” and satisfied [57]. Compared to the few published studies on volunteer civilians, many studies have been conducted on medical students and future healthcare professionals’ volunteering, reporting a sense of giving real aid among the benefits, driven by altruism and the ethical imperative to serve their community [58,59,60,61,62].

Concerning training needs, although three-quarters of our volunteers had already acquired several certifications on specific skills, they displayed a good interest to increase their level of knowledge and training, mainly in specialized volunteering activities and in the field of strategies to cope with stress and manage anxiety. The scientific literature has often debated the actual adequacy of the training level of “novice” volunteers and the possible greater impact that stressors could have on them [35]. In our sample, our “novice” and less experienced volunteers did not seem to confirm this outcome, confirming instead the “cumulative stress” concept often experienced by those who have been involved in many different emergencies and have a wide experience of volunteering behind them [63,64].

The voluntary sector is extremely important both to the general public and to the environment. Following the suggestions of Dolce and Ricciardi [35], to improve volunteers training on technical and emotional skills, the Abruzzo Civil Protection associations, PIVEC, NOVPC, and LARES, signed a research and training agreement with the university to allow volunteers involved in emergency services for a long time to report, if present, their levels of post-traumatic distress and to plan specific training to improve the utilization of their emotional resources. The project is still ongoing. Together with the involved Civil Protection associations, it will plan targeted interventions, centered on their expressed needs (not necessarily psychological ones), more often tied to improve their technical competence in facing disasters and increase their sense of competence and mastery.

## 6. Strengths and Limitations

To the best of our knowledge, so far, the present study is one of the few studies in the literature to examine the variables previously described in a sample of Civil Protection volunteers and the first Italian study during the first wave of the COVID-19 emergency.

Nevertheless, this study presents some limitations. The first and the main limitation concerns the limited size of the sample. Only a little more than 10% of volunteers filled in the online assessment, despite their belonging associations being interested in offering the opportunity to assess their eventual emotional distress and to offer support whenever it would be requested. These data are not different from those of Dolce and Ricciardi [29] where, for describing the impact of psychological risk factors on disaster rescue of Italian Civil Protection volunteers, had a response rate of 7.8%.

Several reasons were hypothesized to explain the limited adhesion to the survey: the difficulty for the volunteers of showing “fragility” on a psychological level with the fear of compromising their self-image, the lack of time to fill in the questionnaires, and the overall time duration of the online assessment. Other studies on emergency workers have repeatedly confirmed the current professional culture in which volunteers may be generally reluctant to recognize the symptoms of psychological distress, with the fear that it is interpreted as a sign of weakness and inadequacy that would make them seem unable to do their jobs [51]. Therefore, the subtle underlying prejudice that, although anonymous, acceptance to undergo a mental health assessment could itself be a sign of “weakness” could be a further possible explanation of the lack of participation in the survey of the Civil Protection volunteers.

## 7. Conclusions

The level of anxious, depressive, and post-traumatic symptoms of volunteering was very low and did not meet the threshold for clinical relevance in the studied group, which showed good emotional conditions and a high utilization of problem-focused coping strategies. The “satisfaction” to help in a “new” distressing situation, such as the COVID-19 pandemic, especially shown by less experienced volunteers, could have overwhelmed their emotional difficulties. We suppose that, among others, the pro-social behaviors of the Civil Protection volunteers could have been addressed by strong motivation and a sense of belonging the L’Aquila community, which has already suffered a devastating earthquake, and for which emotional and materials wounds are still open.

We cannot always and only rely on motivation and commitment toward the community. The voluntary sector, extremely important both to the general population and to the environment, needs attention, and new “urban” emergencies will give us the opportunities to further study and improve both the technical and emotional skills of our volunteers.

## Figures and Tables

**Figure 1 ijerph-18-08587-f001:**
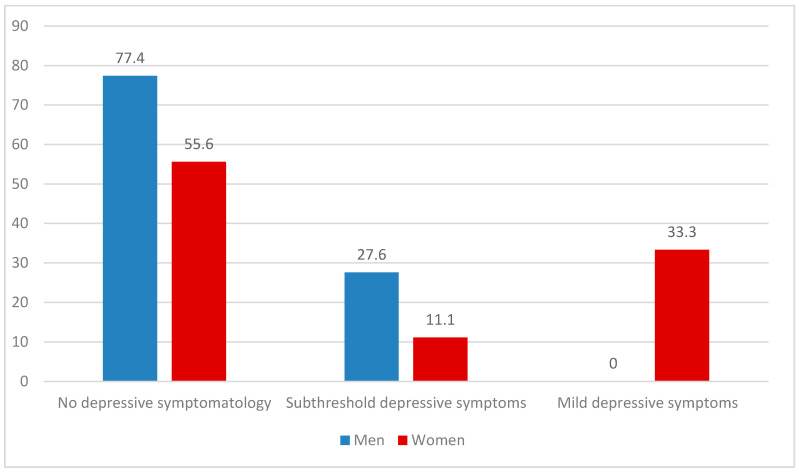
Depressive symptomatology (as measured by PHQ-9) reported by volunteers by gender (%).

**Figure 2 ijerph-18-08587-f002:**
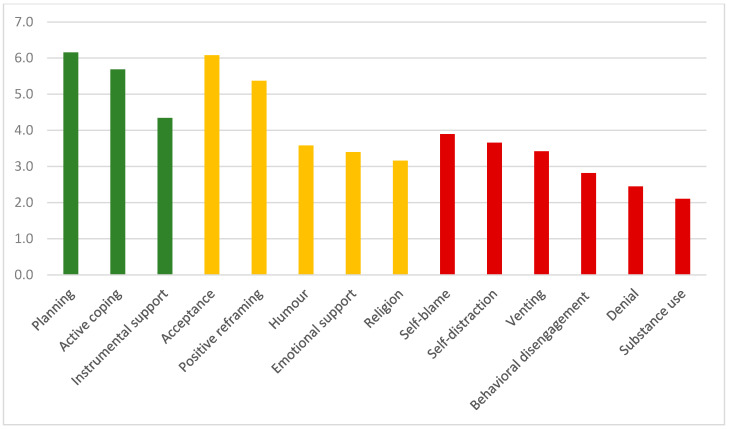
Brief-COPE scores of the 14 coping styles referred to the 3 composite dimensions measuring problem-focused strategies (green bars), emotion-focused strategies (yellow bars), and dysfunctional coping strategies (red bars) (entire sample of 38 volunteers).

**Table 1 ijerph-18-08587-t001:** Items of the survey’s second section related to the Civil Protection volunteer experiences.

**Item 1**	Was the COVID-19 pandemic your first volunteer experience?
**Item 2**	How many years have you been a Civil Protection volunteer?
**Item 3**	Where did you operate?
**Item 4**	In what emergencies did you give your contribution?
**Item 5**	About the emergency COVID-19 as a volunteer, which function/task have you done?
**Item 6**	About the COVID-19 emergency as a volunteer, how many days have you been engaged?
**Item 7**	How much did volunteering in the COVID-19 emergency interfere with your work and/or study activities?
**Item 8**	If you participated in volunteer activities for the COVID-19 emergency, would you like to share your recent experiences with someone?
**Item 9**	As a Civil Protection volunteer, did you follow specific training courses?
**Item 10**	In the future, which specific training courses would you like to participate in?

**Table 2 ijerph-18-08587-t002:** Socio-demographic characteristics of the sample.

Characteristics	Volunteers Serving during the COVID-19 Pandemic (*n* = 30)	Volunteers Non-Serving during the COVID-19 Pandemic (*n* = 8)
**Gender (%)**		
Male	23 (76.7)	6 (75)
Female	7 (23.3)	2 (25)
**Age, M (SD)**	35.9 (15.1)	43.6 (6.8)
**Age ranges (%)**		
16–29 years	13 (43.3)	--
30–49 years	11 (36.7)	6 (75)
50 years and above	6 (20)	2 (25)
**Education (%)**		
Primary school	1 (3.3)	--
Secondary school	11 (36.7)	--
High school	16 (53.3)	7 (87.5)
Graduated	1 (3.3)	1 (12.5)
PhD	1 (3.3)	--
**Marital status (%)**		
Single	12 (40)	5 (62.5)
Married	15 (50)	1 (12.5)
Divorced	2 (6.7)	2 (25.0)
Widow/widower	1 (3.3)	--
**Working conditions (%)**		
Unemployed	4 (13.3)	--
Full-time work	13 (43.3)	7 (87.5)
Part-time work (waiters, caregiver of disabled, cooperative member, maid)	5 (16.7)	1 (12.5)
Student	6 (20)	--
Retired from work	2 (6.7)	--

**Table 3 ijerph-18-08587-t003:** Experience in the voluntary sector of the sample (*n* = 38).

Variables	Sample of Volunteers
**Years of volunteer service, M (SD)**	6.5 (7.9)
**Years of volunteer service (%)**	
<1 year	11 (28.9)
1 to 9 years	18 (47.4)
10 to 20 years	6 (15.8)
>20 years	3 (7.9)
**First experience as a volunteer during the COVID-19 pandemic (%)**	
Yes	13 (34.2)
No	25 (65.8)
**Territories of volunteer experiences (%) (*n* = 25)**	
In region of Abruzzo	13 (52)
In the Italian territories	10 (40)
Outside Italy	2 (8)
**Type of emergency (%) (*n* = 25)**	
Earthquakes	21 (84)
Fires	20 (80)
Floods	14 (66.6)
Avalanches	4 (16)
Temperature extremes/ Snow emergencies	3 (12)
**Service during the COVID-19 emergency (%)**	
No service	8 (21.1)
<1 month	5 (13.2)
2–3 months	12 (31.3)
4–6 months	13 (34.4)
**Activities conducted during the COVID-19 emergency (%) (*n* = 30)**	
Organizational, secretarial, and monitoring interventions	15 (50)
Activities “on the field” (driving ambulances, transporting food, distributing individual protection disposals)	15 (50)
**Impact of volunteering on private work/study (%) (*n* = 30)**	
No impact	10 (33.3)
Occasional difficulties	1 (3.3)
Some difficulties	11 (36.7)
Many difficulties	4 (13.3)
Major difficulties	4 (13.3)
**Willing to share experiences of volunteering during the COVID-19 emergency with (%) (*n* = 30)**	
Friends/colleagues	17 (56.7)
A professional	9 (30)
No need to talk about/no answer	4 (13.3)
**Certification of specific training courses (%)**	
Yes	28 (73.7)
No	10 (26.3)
**Training needs (%)**	
Courses related to specific natural disaster	21 (55.3)
Training to better manage distress/anxiety	7 (18.4)
Information technology courses	2 (5.3)
Courses for driving specific vehicles (bobcat)	1 (2.6)
Missing	7 (18.4)

**Table 4 ijerph-18-08587-t004:** Psychopathological and coping strategies data of Civil Protection volunteers serving during the COVID-19 pandemic compared to those non-serving during the COVID-19 pandemic.

Measures	Volunteer Serving during the COVID-19 Pandemic (*n* = 30)	Volunteers non-Serving during the COVID-19 Pandemic (*n* = 8)	
**GHQ-12 total score, M (SD)**	8.3 (4.6)	8.6 (6.04)	F = 0.030; *p* = 0.864
normal (%)—range 0–14	90	87.5	
moderate psychological distress (%)—range 15–19	6.7	12.5	
severe psychological distress (%)—range 20–36	3.3	--	
**PHQ-9 total score, M (SD)**	3.9 (3.3)	4.0 (3.9)	F = 0.005; *p* = 0.943
No depressive symptoms (%)—range 1–4	70	62.5	
Subthreshold depressive symptoms (%)—range 5–9	23.3	25.0	
Mild depressive symptoms (%)—range 10–14	6.7	12.5	
**PCL-5 score 4 diagnostic criteria of the DSM 5, M (SD)**	9.3 (10.0)	8.1 (9.7)	F = 0.098; *p* = 0.756
B. Intrusiveness	0.74 (0.79)	0.47 (0.64)	F = 0.798; *p* = 0.378
C. Avoidance	0.46 (0.82)	0.37 (0.44)	F = 0.090; *p* = 0.767
D. Negative cognitions and mood	0.31 (0.45)	0.44 (0.59)	F = 0.461; *p* = 0.501
E. Hyperarousal	0.41 (0.43)	0.31 (0.43)	F = 0.369; *p* = 0.547
**Brief–COPE dimensions, M (SD)**			
Problem-focused strategies	5.5 (1.2)	4.6 (0.9)	F = 3.772; *p* = 0.060
Emotion-focused strategies	4.3 (1.1)	4.0 (0.8)	F = 0.563; *p* = 0.458
Dysfunctional coping strategies	3.0 (0.8)	2.9 (0.8)	F = 0.084; *p* = 0.773

**Table 5 ijerph-18-08587-t005:** Psychopathological and coping strategies data of Civil Protection less experienced volunteers serving during the COVID-19 pandemic compared to more experienced volunteers.

Measures	Less Experienced Volunteers (Less than 10 Years of Service) (*n* = 22)	More Experienced Volunteers (10 Years or More of Service) (*n* = 8)	
**GHQ-12 total score, M (SD)**	7.0 (3.0)	11.7 (5.6)	F = 8.688; *p* = 0.006
**PHQ-9 total score, M (SD)**	3.0 (2.3)	6.2 (4.6)	F = 6.191; *p* = 0.019
**PCL-5 total score (SD)**	7.4 (8.0)	13.3 (4.7)	F = 3.232; *p* = 0.083
**PCL-5 score 4 diagnostic criteria of the DSM 5, M (SD)**			
B. Intrusiveness	0.65 (0.73)	1.00 (0.93)	F = 1.123; *p* = 0.298
C. Avoidance	0.38 (0.75)	0.68 (1.03)	F = 0.766; *p* = 0.389
D. Negative cognitions and mood	0.20 (0.30)	0.62 (0.66)	F = 5.846; *p* = 0.022
E. Hyperarousal	0.33 (0.32)	0.64 (0.60)	F = 3.341; *p* = 0.078
**Brief–COPE dimensions, M (SD)**			
Problem-focused strategies	5.6 (1.2)	5.5 (1.3)	F = 0.015; *p* = 0.913
Emotion-focused strategies	4.4 (1.0)	4.2 (1.2)	F = 0.158; *p* = 0.694
Dysfunctional coping strategies	3.0 (0.7)	3.1 (1.1)	F = 0.010; *p* = 0.922

**Table 6 ijerph-18-08587-t006:** Coping scale data of Civil Protection volunteers serving during the COVID-19 pandemic compared to those non-serving during the COVID-19 pandemic.

Measures	Group	*n*	Mean Rank	*U*	Z	*p*
1. Positive reframing	Volunteer serving during the COVID-19 pandemic	30	19.07	107.00	−0.477	0.661
	Volunteers non-serving during the COVID-19 pandemic	8	21.13			
2. Self-distraction	Volunteer serving during the COVID-19 pandemic	30	19.85	109.50	−0.385	0.712
	Volunteers non-serving during the COVID-19 pandemic	8	18.19			
3. Venting	Volunteer serving during the COVID-19 pandemic	30	19.65	115.50	−0.168	0.875
	Volunteers non-serving during the COVID-19 pandemic	8	18.94			
4. Use of instrumental support	Volunteer serving during the COVID-19 pandemic	30	19.97	106.00	−0.529	0.635
	Volunteers non-serving during the COVID-19 pandemic	8	17.75			
5. Active coping	Volunteer serving during the COVID-19 pandemic	30	21.32	65.50	−2.002	0.045
	Volunteers non-serving during the COVID-19 pandemic	8	12.69			
6. Denial	Volunteer serving during the COVID-19 pandemic	30	18.85	100.50	−0.938	0.492
	Volunteers non-serving during the COVID-19 pandemic	8	21.94			
7. Religion	Volunteer serving during the COVID-19 pandemic	30	21.08	72.50	−1.911	0.056
	Volunteers non-serving during the COVID-19 pandemic	8	13.56			
8. Humour	Volunteer serving during the COVID-19 pandemic	30	19.07	107.00	−0.479	0.661
	Volunteers non-serving during the COVID-19 pandemic	8	21.13			
9. Behavioral disengagement	Volunteer serving during the COVID-19 pandemic	30	19.20	111.00	−0.367	0.765
	Volunteers non-serving during the COVID-19 pandemic	8	20.63			
10. Use of emotional support	Volunteer serving during the COVID-19 pandemic	30	19.02	105.50	−0.540	0.610
	Volunteers non-serving during the COVID-19 pandemic	8	21.31			
11. Substance use	Volunteer serving during the COVID-19 pandemic	30	19.90	108.00	−0.919	0.686
	Volunteers non-serving during the COVID-19 pandemic	8	18.00			
12. Acceptance	Volunteer serving during the COVID-19 pandemic	30	21.38	63.500	−2.068	0.041
	Volunteers non-serving during the COVID-19 pandemic	8	12.44			
13. Planning	Volunteer serving during the COVID-19 pandemic	30	21.07	73.00	−1.738	0.082
	Volunteers non-serving during the COVID-19 pandemic	8	13.63			
14. Self-blame	Volunteer serving during the COVID-19 pandemic	30	19.78	111.50	−0.314	0.765
	Volunteers non-serving during the COVID-19 pandemic	8	18.44			

**Table 7 ijerph-18-08587-t007:** Coping scale data of Civil Protection less experienced volunteers serving during the COVID-19 pandemic compared to more experienced volunteers.

Measures	Group	*n*	Mean Rank	*U*	Z	*p*
1. Positive reframing	Less experienced volunteers (<10 years of service)	22	15.84	80.50	−0.359	0.730
	More experienced volunteers (10 years or more of service)	8	14.56			
2. Self-distraction	Less experienced volunteers (<10 years of service)	22	15.98	77.50	−0.506	0.629
	More experienced volunteers (10 years or more of service)	8	14.19			
3. Venting	Less experienced volunteers (<10 years of service)	22	14.73	71.00	−0.840	0.447
	More experienced volunteers (10 years or more of service)	8	17.63			
4. Use of instrumental support	Less experienced volunteers (<10 years of service)	22	16.39	68.50	−0.968	0.368
	More experienced volunteers (10 years or more of service)	8	13.06			
5. Active coping	Less experienced volunteers (<10 years of service)	22	14.82	73.00	−0.724	0.504
	More experienced volunteers (10 years or more of service)	8	17.38			
6. Denial	Less experienced volunteers (<10 years of service)	22	14.61	68.50	−1.311	0.368
	More experienced volunteers (10 years or more of service)	8	17.94			
7. Religion	Less experienced volunteers (<10 years of service)	22	16.77	60.00	−1.421	0.202
	More experienced volunteers (10 years or more of service)	8	12.00			
8. Humour	Less experienced volunteers (<10 years of service)	22	15.36	85.00	−0.146	0.909
	More experienced volunteers (10 years or more of service)	8	15.88			
9. Behavioral disengagement	Less experienced volunteers (<10 years of service)	22	16.11	74.50	−0.735	0.534
	More experienced volunteers (10 years or more of service)	8	13.81			
10. Use of emotional support	Less experienced volunteers (<10 years of service)	22	15.07	78.50	−0.465	0.662
	More experienced volunteers (10 years or more of service)	8	16.69			
11. Substance use	Less experienced volunteers (<10 years of service)	22	16.05	76.00	−1.081	0.597
	More experienced volunteers (10 years or more of service)	8	14.00			
12. Acceptance	Less experienced volunteers (<10 years of service)	22	15.64	85.00	−0.145	0.909
	More experienced volunteers (10 years or more of service)	8	15.13			
13. Planning	Less experienced volunteers (<10 years of service)	22	15.64	85.00	−0.145	0.909
	More experienced volunteers (10 years or more of service)	8	15.13			
14. Self-blame	Less experienced volunteers (<10 years of service)	22	15.45	87.00	−0.048	0.982
	More experienced volunteers (10 years or more of service)	8	15.63			

**Table 8 ijerph-18-08587-t008:** Correlations among the age of volunteers, GHQ-12, PHQ-9 total scores, the 4 dimensions (B-E criteria of DSM-5) of the PCL5, and the 3 Brief-Cope coping styles identified by “Problem-focused strategies”, “Emotion-focused strategies” and “Dysfunctional coping strategies” (*n* = 30).

Measures		Age	GHQ12 Total Score	PHQ-9 Total Score	PCL5 B. Intrusiveness	PCL5 C. Avoidance	PCL5 D. Negative Cognitions and Mood	PCL5 E. Hyperarousal	BC Problem-Focused Strategies	BC Emotion-Focused Strategies
**GHQ-12 total score**	Pearson’s Correlation	0.032	--							
	2-tailed *p*-value	0.865								
**PHQ-9 total score**	Pearson’s Correlation	0.000	0.728 **	--						
	2-tailed *p*-value	0.998	0.000							
**PCL5 B. Intrusiveness**	Pearson’s Correlation	0.134	0.382 *	0.551 *	--					
	2-tailed *p*-value	0.482	0.037	0.002						
**PCL5** **C. Avoidance**	Pearson’s Correlation	0.338	0.275	0.416 *	0.880 **	--				
	2-tailed *p*-value	0.067	0.141	0.022	0.000					
**PCL5 D. Negative cognitions and mood**	Pearson’s Correlation	0.263	0.586 **	0.712 **	0.754 **	0.747 **	--			
	2-tailed *p*-value	0.161	0.001	0.000	0.000	0.000				
**PCL5 E. Hyperarousal**	Pearson’s Correlation	0.194	0.622 **	0.704 **	0.493 **	0.442 *	0.761 **	--		
	2-tailed *p*-value	0.304	0.000	0.000	0.006	0.014	0.000			
**BC Problem-focused strategies**	Pearson’s Correlation	0.062	−0.051	−0.100	0.033	0.070	−0.019	0.070	--	
	2-tailed *p*-value	0.746	0.790	0.598	0.862	0.714	0.921	0.714		
**BC Emotion-focused strategies**	Pearson’s Correlation	0.149	−0.091	0.189	0.385 *	0.440 *	0.415 *	0.197	0.532 **	--
	2-tailed *p*-value	0.431	0.633	0.318	0.036	0.015	0.023	0.297	0.002	
**BC dysfunctional coping strategies**	Pearson’s Correlation	0.146	0.491 **	0.587 **	0.686 **	0.654 **	0.725 **	0.570 **	0.168	0.564 **
	2-tailed *p*-value	0.441	0.006	0.001	0.000	0.000	0.000	0.001	0.374	0.001

** *p* < 0.01; * *p* < 0.05.

## Data Availability

The data presented in this study are available on request from the corresponding author. The data are not publicly available due to the agreement established with the Civil Protection organizations involved in the study.

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
