# Peer review of "“Hang in There!”: Mental Health in a Sample of the Italian Civil Protection Volunteers during the COVID-19 Health Emergency"

_ijerph, 2021, doi:10.3390/ijerph18168587_

Round 1
Reviewer 1 Report
Is an interesting workpiece, worth to be read. The paper offers a complete picture of a very specific population, with a clear indication of strengths and limitations of the study.

Author Response
We thank the reviewer for the appreciation of our paper and for understanding the limitations of the study.
Reviewer 1 seems to appreciate the
a) the research questions;
b) the methodology;
c) the presentation of the results;
d) the local impact of the research.
Reviewer 2 Report
overall it is well written. Due to the small sample size of those who not serving during pandemic (n=8), the comparison to those who served during pandemic does not make statistical sense. Suggest to removing it and consistently reporting the results in a purely descriptive way.
Author Response
Thanks for your suggestions.
Due to the small sample size of comparison groups, as suggested by reviewer 3, we utilized a non-parametric test (the Mann–Whitney U test) instead of ANOVA.
Reviewer 3 Report
The manuscript by Roncone et al. entitled “Hang in There!”: the Well-Being in Serving in a Sample of the3 Italian Civil Protection Volunteers of the L’aquila Province, Abruzzo, during the Covid-19 Health Emergency” examines the impact of the COVID-19 pandemic on different facets of mental health (e.g., self-reported depressive symptoms, anxiety) of a sample of Italian volunteers engaged by the Italian Civil Protection. The study is innovative and very interesting, however below I report some aspects that in my opinion must be addressed.
Considering the contents of the article, I think that the title can be shorted and be changed in part. I suggest something like “Hang in There!”: Mental Health in a sample of Italian Civil Protection Volunteers during the Covid-19 Health Emergency. In my opinion, using well-being can be controversial, since the authors administered any psychological/physical well-being questionnaire but a self-report measure of depressive symptoms and one assessing anxiety. Furthermore coping is considered a multidimensional construct that can be associated with well-being but it is not often considered an index of well-being.
The introduction is well structured; however, I do have some concerns and suggestions to improve its relevance.
I think that a more focused analysis of the literature concerning the impact of the emergency events (like the COVID-19 pandemic) on the dimensions of mental health that the authors assessed in the Civil Protection volunteers need to be developed. In this regard, I report below some articles that could be useful to pursue this goal and to justify the study:
- Crescenzo, P., Marciano, R., Maiorino, A., Denicolo, D., D’Ambrosi, D., Ferrara, I., ... & Diodato, F. (2021). First COVID-19 wave in Italy: coping strategies for the prevention and prediction of burnout syndrome (BOS) in voluntary psychologists employed in telesupport. Psychology Hub, 38(1), 31-38.
- Maiorano, T., Vagni, M., Giostra, V., & Pajardi, D. (2020). COVID-19: risk factors and protective role of resilience and coping strategies for emergency stress and secondary trauma in medical staff and emergency workers—an online-based inquiry. Sustainability, 12(21), 9004.
- Milligan-Saville, J., Choi, I., Deady, M., Scott, P., Tan, L., Calvo, R. A., ... & Harvey, S. B. (2018). The impact of trauma exposure on the development of PTSD and psychological distress in a volunteer fire service. Psychiatry Research, 270, 1110-1115.
- Perrin, M. A., DiGrande, L., Wheeler, K., Thorpe, L., Farfel, M., & Brackbill, R. (2007). Differences in PTSD prevalence and associated risk factors among World Trade Center disaster rescue and recovery workers. American Journal of Psychiatry, 164(9), 1385-1394.
- Vagni, M., Maiorano, T., Giostra, V., & Pajardi, D. (2020). Hardiness and coping strategies as mediators of stress and secondary trauma in emergency workers during the COVID-19 pandemic. Sustainability, 12(18), 7561.
- Vujanovic, A. A., Lebeaut, A., & Leonard, S. (2021). Exploring the impact of the COVID-19 pandemic on the mental health of first responders. Cognitive Behaviour Therapy, 50(4), 320-335.
In the introduction, on page 2, the authors state that the Italian Civil Protection “encourages the participation of volunteer organizations in civil activities and emergency responses” (line 47-48). I suggest stressing more about what exactly makes a volunteer of the Italian Civil Protection (just a few examples in parentheses to help the less expert readers would be sufficient).
Page 2, lines 61-63: since PTSD is a serious menace for the volunteers, I suggest describing shortly what this disorder is and its symptoms.
Page 2, line 69: it would be useful to the less expert reader a clarification with some examples of the “psychopathological signs” mentioned by the authors.
Page 2, line 74: Positive coping strategies need to be defined since there are many types of them. Furthermore, it is crucial since the authors administered a coping questionnaire in their study.
Page 2, line 77 what are emotional distress symptoms? It is sufficient that the authors provide some examples in parentheses.
Page 3, lines 116-117. Please, provide an operational definition of psychological conditions and psychopathological health conditions to point out their differences.
The authors proposed only 1 hypothesis, but they stated that the aims of the study are three. The authors must specify better the first goal that in my opinion is too generic and they should specify better the hypotheses justifying them based on the existing literature. For instance, it is not clear what the authors expect relatively to the coping strategies. In this part of the article, they did not specify what sort of coping strategy was examined in their study and what their hypothesis is.
Paragraph 2.2 must be reported before the presentation of the study since it can be considered part of the introduction to the study.
Page 5: please report the internal consistency of each questionnaire.
Page 5, paragraph 2.3.3: please provide an operational definition of problem-focused strategies, emotion-focused strategies, and dysfunctional coping strategies.
My major concerning this study is the choice of some statistical analyses. Specifically, the authors don’t clarify whether the data distribution was explored and whether they found the normality of their data, or whether they use some correction to normalize their data. Even though the distribution of their data was normal, the authors must consider that their sample is very limited and unbalanced in terms of frequency of volunteers engaged during the COVID-19 pandemic (n = 30) or not engaged (n = 8). In my opinion, having only 11% of the respondents is a serious limitation of the study that must be very cautious in generalizing the findings to the volunteers engaged during an emergency event.
Based on all these considerations, the application of ANOVA is not appropriate. I strongly suggest using non-parametric tests instead of ANOVA to compare the two groups of volunteers and to compare more experienced (> 10 years vs. less 10 years of experience in the Civil Protection) and examine the effect of gender in the different mental health conditions.
Paragraph 4.1.: The authors state that they did not find any difference when they compared two groups of volunteers differently matched (e.g., very expert volunteer with more than 10 years of experience vs. less expert volunteer). However, in the paragraph, the authors often argue that they didn’t find any significant difference (e.g., “No statistically significant difference was found in age and level of education by gender”) but they often don’t report any statistic. This is not sufficient; they must report the test and the values. Moreover, I think that considering the aims of the study, it is more appropriate that the authors use some non-parametric tests to explore the differences between the volunteers engaged during the pandemic and those not involved in the emergency event (e.g., age, education, years of experience like volunteer, marital status, etc.).
Paragraph 4.2: in the main text the authors do report only ‘no statistical differences’ were found. Considering that the ANOVAs are not appropriate, reporting the non-parametric comparisons, the authors must report all analyses that they performed and the statistical values in the main text and then summarize the most relevant information in Tables. Furthermore, when a significant effect is reported, the authors must also report the effect size.
I suggest including ‘years of volunteers service’ in the correlational analysis controlling for the effect of age (i.e., performing partial correlations instead of simple Pearson’s correlations).
I don’t have understood what analysis was conducted to examine the third aim of the study (i.e., assess the training needs of the volunteers).
In the discussion, the results emerging from the appropriate statistical analyses must be discussed considering both the existing literature and better-specified hypotheses. Moreover, considering the paucity of respondents caution is needed in generalizing the findings. Therefore, at present, I can’t assess the quality of the discussion section since the hypotheses must be clarified and the ANOVAs are not appropriate and must be substituted with some non-parametric (and more robust) tests.
Future research directions should also be pointed out. Moreover the section of the limits must be developed, highlighting further limits of the study (e.g., gender not balanced, small battery of tests, years of service not balanced etc).
Author Response
Thanks for your suggestions, which allowed us to improve our paper. We provided the suggested changes.
We shorted the title as suggested, “Hang in There!”: Mental Health in a sample of Italian Civil Protection Volunteers during the Covid-19 Health Emergency.
We included the suggested references in our work.
page 2, lines 47-48
In the introduction, we included a series of activities of the Italian Civil Protection volunteer organizations.
page 2, lines 61-63
We shortly described the PTSD (DSM 5, APA 2013).
page 2, lines 69
We corrected the sentence: “However, research has found that potentially traumatic experiences do not necessarily lead to the development of psychopathological symptoms”.
page 2, line 74
We reported the definition of positive coping strategies.
page 2, line 77
We reported examples of emotional distress symptoms.
page 3 lines 116-117
We corrected the sentence: “The aims of the study were: 1) describe the psychopathological health conditions, and the different coping styles (.... ) adopted by the volunteers' sample; 2) examine the correlations among emotional distress, psychopathological symptoms, and individual coping strategies; 3) assess the training needs of the volunteers.”
We improved the specific aims and the hypotheses of the work.
Paragraph 2.2 was reported before the presentation of the study.
Page 5: We reported the internal consistency of each questionnaire.
Page 5, paragraph 2.3.3
We defined problem-focused strategies, emotion-focused strategies, and dysfunctional coping strategies
***
We agree with Reviewer 3 about the problem of statistical analysis in our small sample of subjects.
We used a non-parametric test (instead of ANOVA, we used Mann-Whitney u test) and for significant effects, and we reported the effect size.
We revised Tables 2, 4, 5.
To justify the results in the text we added Table 6-7.
As suggested, we performed a partial correlation, including ‘years of volunteers service’, controlled for ‘age’ (Table 8).
Concerning the third aim, we better described the results (we added paragraph 4.5) concerning the training needs of volunteers.
Based on your suggestions, the discussion was improved (the results do not differ so much from those reported in our previous version).
The future research direction of our project was already briefly reported.
About the limitations of the study, we do not agree with your suggestions. Based on national data ISTAT 2013 (Reference 1), women are less represented compared to men in organized volunteering activities (men rate 13.3%, women rate 11.9% on 100). The online test battery was large enough, or better, it was too large! Volunteers complained about the length of the survey (the mean time for filling the survey was 25 minutes)! About the years not balanced, in this study, we used a convenience sample.
Round 2
Reviewer 3 Report
The revisions made by the authors are satisfactory according to the comments made.